# Sexual selection gradients change over time in a simultaneous hermaphrodite

Jeroen NA Hoffer, Janine Mariën, Jacintha Ellers, Joris M Koene*

Section of Animal Ecology, Department of Ecological Science, Faculty of Earth and Life Sciences, Vrije Universiteit, Amsterdam, The Netherlands

**Abstract** Sexual selection is generally predicted to act more strongly on males than on females. The Darwin-Bateman paradigm predicts that this should also hold for hermaphrodites. However, measuring this strength of selection is less straightforward when both sexual functions are performed throughout the organism's lifetime. Besides, quantifications of sexual selection are usually done during a short time window, while many animals store sperm and are long-lived. To explore whether the chosen time frame affects estimated measures of sexual selection, we recorded mating success and reproductive success over time, using a simultaneous hermaphrodite. Our results show that male sexual selection gradients are consistently positive. However, an individual's female mating success seems to negatively affect its own male reproductive success, an effect that only becomes visible several weeks into the experiment, highlighting that the time frame is crucial for the quantification and interpretation of sexual selection measures, an insight that applies to any iteroparous mating system.

*For correspondence: joris.
koene@vu.nl

Reviewing editor: David
Lentink, Stanford University,
United States

## Introduction

Darwin defined sexual selection as selection on traits that affect mating success (*Darwin, 1871*). In doing so, he clearly focused on the obvious secondary sexual characters that often differ between males and females. Classical examples include antlers of deer, long and extravagantly coloured (tail) feathers in birds and traits of that ilk. In recent decades, this definition has been refined, most notable due to the realisation that sexual selection does not only act prior to mating – referred to as pre-copulatory sexual selection (Darwin's focus) - but also after mating - post-copulatory sexual selection (e.g., *Parker, 1970*; *Eberhard, 1996*). Investigating pre-copulatory sexual selection involves measuring mating success, which is generally defined as the number of mates that an individual copulates with (for males $MS_m$; for females $MS_f$), and the resulting reproductive success which is measured as the number of offspring that an individual is able to produce (for males $RS_m$; for females $RS_f$). The residual variation of the relationship between mating success and reproductive success can then be used as a quantitative proxy for post-copulatory sexual selection. These sexual selection processes are often measured, but with a strong bias towards species with separate sexes, even though simultaneous hermaphrodites are under influence of the same selective pressures (*Charnov, 1979*; *Nakadera and Koene, 2013*; *Pélissié et al., 2012*; *Schärer and Pen, 2013*).

One, now classical, study on *Drosophila melanogaster* by *Bateman (1948)* sparked the pivotal insight that the factors that limit reproductive success for males and females are different. This is referred to as the Darwin-Bateman paradigm (or Bateman principle; reviewed in *Dewsbury, 2005*). Due to anisogamy, for which sexual selection has been an important driving force (reviewed in *Parker and Birkhead, 2013*), a clear difference is found in the cost of producing male and female gametes. As a consequence, sperm production is generally not a limiting factor for males, so the number of fathered offspring (male reproductive success, $RS_m$; *Bateman, 1948*) depends directly on the number of mates the male can inseminate. In contrast, egg production is highly dependent on

**eLife digest** Many factors affect an organism's ability to survive and reproduce. These factors are often called "selection pressures" and include the availability of food and shelter, conditions in the environment such as temperature, and the presence of diseases and predators. Males and females experience different selection pressures so they often evolve to look different – consider, for example, the male deer's antlers and the peacock's colourful tail feathers. Such traits arise from a phenomenon called sexual selection, the selection pressures that act on an organism's ability to obtain a mate.

Measuring sexual selection is not only of interest to scientists looking to understand how evolutionary processes work; it also has wider applications, including in wildlife conservation. For instance, knowing which cues are important for successful reproduction could help efforts to breed endangered animals in captivity and stop them from going extinct.

Scientists study sexual selection in a species by measuring how successful males and females are at mating and reproducing. Past studies have found that a female's reproductive success mainly depends on there being enough resources available for her to produce eggs, while a male's success depends on him getting access to these eggs. However, most research into sexual selection has been on species with separate sexes. It is more difficult to measure sexual selection in species – like snails and slugs – where each individual is male and female at the same time. As such, it is not clear if reproductive success in these species, which are known as simultaneous hermaphrodites, depends on the same factors as those species with separate sexes.

To address this, Hoffer et al. measured sexual selection in the great pond snail *Lymnaea stagnalis*, a simultaneous hermaphrodite. Most studies estimate sexual selection based on measurements taken over several days. Instead, Hoffer et al. observed the great pond snail over a period of eight weeks, which is about a quarter of its reproductive life.

The experiments showed that mating multiple times, especially with multiple partners, overall improves the development of the snail's offspring. The male part of the great pond snail gains the most reproductive success from repeated mating, whereas the female part may in fact be negatively affected. These negative effects were only seen several weeks into the experiment, and so they show that sexual selection pressures change over time.

Future research is needed to determine what causes the negative effects on the female part of the great pond snail. Overall, these findings stress the need for careful consideration of the time frame over which future measurements of sexual selection take place, not just in hermaphrodites, but in all species.

available resources and therefore limits the number of offspring the female can produce (i.e., female reproductive success, RS$_f$; *Bateman, 1948*).

Even though Bateman's experiment has been criticized based on experimental design as well as data collection and actual repeatability (e.g., *Gowaty et al., 2012*; *Gowaty et al., 2013*), the basics of the Darwin-Bateman paradigm still hold (e.g., *Janicke et al., 2016*). Therefore, this paradigm has formed the basis for more formalized approaches to measuring and quantifying sexual selection, in which the difference in variance in mating success and reproductive success of males and females can be captured in different measures of sexual selection (e.g., *Arnold, 1994*; *Jones, 2009*; see Materials and methods). One insightful measure that has emerged is the sexual selection gradient (also referred to as Bateman gradient), which looks at the linear relationship between mating success and reproductive success ($\beta$; e.g., *Arnold and Duvall, 1994*; *Anthes et al., 2010*). The steepness of the male and female Bateman gradient can inform about the strength of sexual selection, as has for example been done for polygamous red jungle fowl (*Collet et al., 2012*), polyandrous rough skinned newts (*Jones, 2009*), and polygynous, role-reversed pipefish (*Jones et al., 2005*).

For species with separate sexes, the above approach is relatively straight forward, because one 'only' needs to regress mating success against reproductive success for each of the sexes and compare the slopes ($\beta_{male}$ and $\beta_{female}$). As pointed out by *Gowaty et al. (2012)*, in order to obtain correct quantifications of sexual selection, independent measures of mating success and reproductive

success are needed, which lacked in Bateman's original study. As furthermore pointed out by *Anthes et al. (2010)*, the quantification of sexual selection is less clear-cut in simultaneous hermaphrodites, because each individual is both male and female at the same time. Hence, besides that mating success in the male role can affect the individual's male reproductive success, it can also directly affect its female reproductive success and vice versa. These interactions between the sexual functions of the individual are referred to as cross-sex effects ($\beta_{mf}$ and $\beta_{fm}$; *Anthes et al., 2010*) and trigger the legitimate question whether these cross-sex effects cause simultaneous hermaphrodites to deviate from the Darwin-Bateman paradigm.

As illustrated by several recent studies, it is possible to quantify sexual selection in simultaneous hermaphrodites (*Ophryotrocha diadema*: *Lorenzi and Sella, 2008*; *Biomphalaria glabrata*: *Anthes et al., 2010*; *Physa acuta*: *Pélissié et al., 2012*; *Macrostomum lignano*: *Marie-Orleach et al., 2016*), with the latter three also applying the sexual selection gradients approach outlined by *Anthes et al. (2010)*. Like in most of the sexual selection gradient studies on separate sexed species, these studies used a very restricted time window (days) within which the strength of sexual selection was estimated. However, many species are long-lived, mate many times and can store and use sperm for extended periods (e.g., *Nakadera and Koene, 2013*; *Nakadera et al., 2014a*). Therefore, the relationship between mating success and reproductive success can be expected to change over time, especially when considering that sperm storage becomes important as soon as individuals are no longer virgin, meaning that the degree to which mating success translated into reproductive success might change (*Baena and Macías-Ordóñez, 2012*; *Wacker et al., 2014*; *Anthes et al., 2016*). Here, we present an experiment in which we quantify mating success and reproductive success, using the great pond snail *Lymnaea stagnalis*, over an eight week period that represents roughly a quarter of its reproductive life in nature.

Our quantification of sexual selection gradients allows us to address several unresolved questions that are of general importance for understanding sexual selection. First, do these simultaneous hermaphrodites conform to the prediction that sexual selection gradients differ for the male and female function? Second, can we detect the predicted cross-sex effects on reproductive success? Third, do sexual selection gradients change depending on the time window of measurement? This latter question, which can be tested given the time frame of our experiment, addresses an issue that has remained experimentally untested in any hermaphroditic species to date (and was only addressed in one separate sexed species: *Turnell and Shaw, 2015*).

In addition, with the collected data, we can address several remaining questions that are specific to the simultaneous hermaphrodite under investigation, the pond snail *L. stagnalis*. We can examine whether partner availability is beneficial for offspring quality, something that is predicted based on the finding that multiple mating results in larger investment per egg in this species (*Hoffer et al., 2012*). Also, by looking at the number of matings in the male and female role, we can determine whether the mating mode of this species is unilateral or relaxed reciprocal (the latter meaning that animals tend to alternate mating roles between successive copulations, possibly with different partners; we already know it is not strict reciprocal: *Koene and Ter Maat, 2005*). As pointed out by *Anthes et al. (2010)*, this information is important, as the mating mode has implications for how independently sexual selection can act on the two sexual roles of hermaphrodites.

## Results

The 200 individuals of the simultaneously hermaphroditic snail *L. stagnalis*, which were all virgin at the start of the experiment, were divided over three treatment groups: Multiple partners, Single partner and No partner (see *Figure 1* and Materials and methods for details). During the whole experiment, a total of 7888 matings were observed in the first two treatments (*Figure 1*). Of the estimated 343,745 eggs that were produced, 1102 were genotyped (excluding eggs from the selfing No partner treatment). At the end of the experiment, 89 egg masses, containing a total of 8407 eggs, were used to determine hatching and developmental success for each treatment. Focals that died during the experiment were excluded from that point onwards from further analysis. At the end of the experiment, a total of 25 out of 125 (20%) snails died in the Multiple partners treatment. For the Single partner and No partner treatments these numbers were 8 out of 46 (17%) and 3 out of 25 (12%), respectively. Overall survival during the whole experiment was 87.8%, which is within the

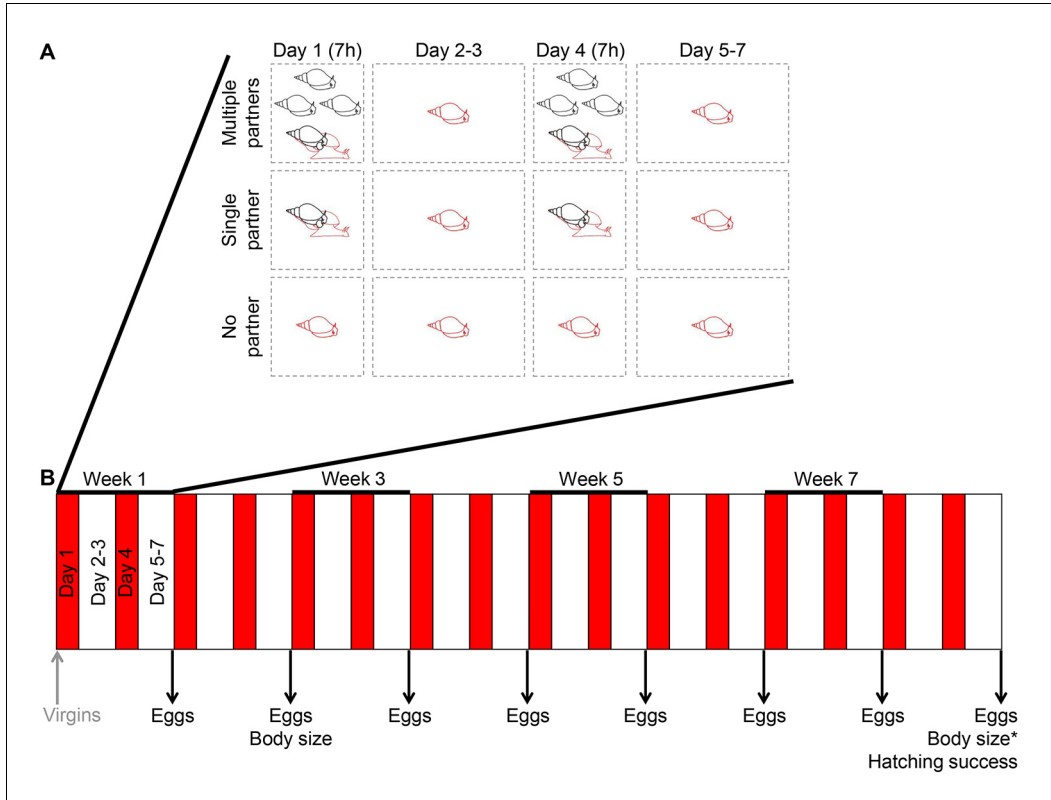

**Figure 1.** The experimental setup used. (**A**) The schematic overview shows the three different treatments that the virgin animals were subjected to for the first week. On Day 1 and 4, animals were allowed to mate for seven hours (7h) with either four different partners (Multiple partners treatment), one partner (Single partner treatment), or no partner (No partner treatment). The focal individual, which had a different microsatellite genotype from its partners, is indicated in red. In between the mating trials, all the animals were kept in isolation, but for simplicity only the focals are shown here. (**B**) The treatments shown in panel A were repeated over the course of the experiment, as indicated by the red bars in the timeline of the eight-week experiment. This also shows that eggs were collected at the end of each experimental week, that body sizes were measured twice, and that a set of eggs that was laid at the end of the experiment was allowed to develop in order to determine hatching success.

common range for this species in our experience, and did not differ between the treatment groups ($\chi^2_2$ = 1.303, p = 0.521).

## Quantification of sexual selection

The measurements of mating activity and reproductive output in the above-mentioned Multiple partners treatment were used to quantify sexual selection by looking at the variance for each sex of one focal within a group of five snails (see Material and methods for details). Due to different limits in terms of their gamete production, variance in the reproductive success of males is expected to be larger than that of females and can be captured in the variance measure $I$. This measure is defined as the standardized variance in relative reproductive success and its value is indicative of the opportunity for selection (***Arnold, 1994***; ***Anthes et al., 2010***; ***Evans and Garcia-Gonzalez, 2016***). The opportunity for sexual selection, which is defined as the standardized variance in relative mating success, is captured in the variance measure $I_s$ (***Arnold, 1994***; ***Anthes et al., 2010***). Comparison of the opportunity for selection values between the male and female role ($I_m$ and $I_f$) shows that these values are larger for the male role (***Figure 2—figure supplement 1*** and ***Figure 2—source data 1***). In addition, these values decreased over the course of the experiment. The much lower values for the opportunity for sexual selection ($I_{sm}$ and $I_{sf}$) reveal a similar trend (***Figure 2—figure supplement 1*** and ***Figure 2—source data 1***).

Subsequently, we compared the sexual selection gradients ($\beta$) between and among the sexual roles. Such a gradient is the linear least-squares regression slope of sex-specific relative RS on sex-specific relative MS (*Jones, 2009*; *Klug et al., 2010*), thus expressing the expected fitness increase achieved by mating one additional time in a specific sex role ($\beta_m$ or $\beta_f$). Because hermaphrodites express both sex functions within one body, the reproductive efforts in one reproductive function can alter reproductive fitness in the other. To deal with this non-independence of male and female reproduction, we used a multiple regression with $MS_m$ and $MS_f$ as explanatory variables on, respectively, $RS_f$ and $RS_m$. As pointed out by *Anthes et al. (2010)*, this approach makes possible cross-sex effects explicit. The resulting cross-sex effects ($\beta_{mf}$ and $\beta_{fm}$) describe how MS in one sex function changes RS in the other sex function.

In order to make the sexual selection gradient measures comparable across time, we used relative values for both mating success and reproductive success; note that time was divided in weeks because eggs were collected at the end of each week (hence the factor Week below). To analyse the sexual selection gradients from the male perspective, we used a model including the dependent variable relative $RS_m$ and the factors relative $MS_m$, relative $MS_f$ and Week, plus their interaction, including focal identity as random, repeated factor. This analysis revealed a significant effect for the factor relative $MS_m$ on relative $RS_m$ ($F_{1, 145.5} = 7.872$; $p = 0.0057$), while Week and the interaction term Week*relative $MS_m$ showed no significance. This indicates that the male sexual selection gradient, $\beta_{mm}$, is positive and does not change over time, as is also clearly reflected in the regression lines (*Figure 2*; see $\beta$-values in *Figure 2—figure supplement 1*, the slopes' confidence intervals in *Figure 2—source data 2*, and slope comparisons between weeks in *Figure 2—source data 3*), revealing that continued mating in the male role assures continuous male reproductive success beyond the first week. When looking at the cross-sex effect $\beta_{mf}$ in this model, the factor relative $MS_f$ had a significant effect on relative $RS_m$ ($F_{1, 145.7} = 29.956$; $p = 0.0001$), which is due to the significant positive relationship between the number of male matings and female reproductive success in the first week, as reflected by the significance of the interaction term Week*relative $MS_f$ ($F_{7, 136.1} = 4.159$; $p = 0.0004$); an effect that disappeared afterwards (*Figure 2* and *Figure 2—figure supplement 1*). From the female perspective, running the same model with relative $RS_f$ as dependent variable, for the female role ($\beta_{ff}$), a significant relationship between female mating success and female reproductive success is found ($F_{1, 140.6} = 5.101$; $p = 0.0255$), which seems to be caused by the negative trend lines found in the last three weeks of the experiment (*Figure 2* and *Figure 2—figure supplement 1*). For the cross sex effect $\beta_{fm}$ the model revealed a significant effect of relative $MS_m$ on relative $RS_f$ ($F_{1, 140,5} = 13.523$; $p = 0.0003$), which seem due to the initial positive correlation (in Week 1 and 2; *Figure 2* and *Figure 2—figure supplement 1*), even though the interaction term Week*relative $MS_m$ was not significant.

Depending on their mating system, male and female mating success may not be fully independent in simultaneous hermaphrodites. Even in unilaterally mating species, playing both roles in a mating encounter in sequence (i.e., reciprocating) could make a multiple regression analysis statistically fragile (*Mitchell-Olds and Shaw, 1987*). To cope with this potential problem, we follow *Anthes et al. (2010)* suggestion to replace $MS_m$ and $MS_f$ by their principal components (PC). When comparing the outcomes of the regression analysis with the principle component analysis (PCA) approach, one can see that most findings were corroborated (comparing *Figures 2* and *3*). For proper interpretation of these graphs, it should be noted that overall mating activity, that is, the correlation component between relative $MS_m$ and relative $MS_f$ is represented by PC2; the sexual bias, that is, the relative difference between $MS_m$ and $MS_f$, is captured by PC1 (see *Figure 3—source data 1* for details; see also *Figure 2—source data 2* for slope comparisons between weeks). In other words, the slope $\beta_{mPC2}$ represents $\beta_{mm}$ and $\beta_{fPC2}$ represents $\beta_{fm}$, and the cross-sex effects are seen in $\beta_{mPC1}$ (= $\beta_{mf}$) and $\beta_{fPC1}$ (= $\beta_{ff}$; *Anthes et al., 2010*). One crucial difference to note between the two analytical approaches is that a more female biased mating rate (i.e., higher PC1 values) can negatively affect male reproductive success (negative slope, $\beta_{mPC1}$), which is not seen in the regression analysis between relative $MS_f$ and relative $RS_m$. Similar negative trend lines are seen for $\beta_{fPC1}$ and $\beta_{ff}$, confirming that a more female biased mating rate may also negatively affect the individual's female reproductive success (in terms of offspring number, but see below).

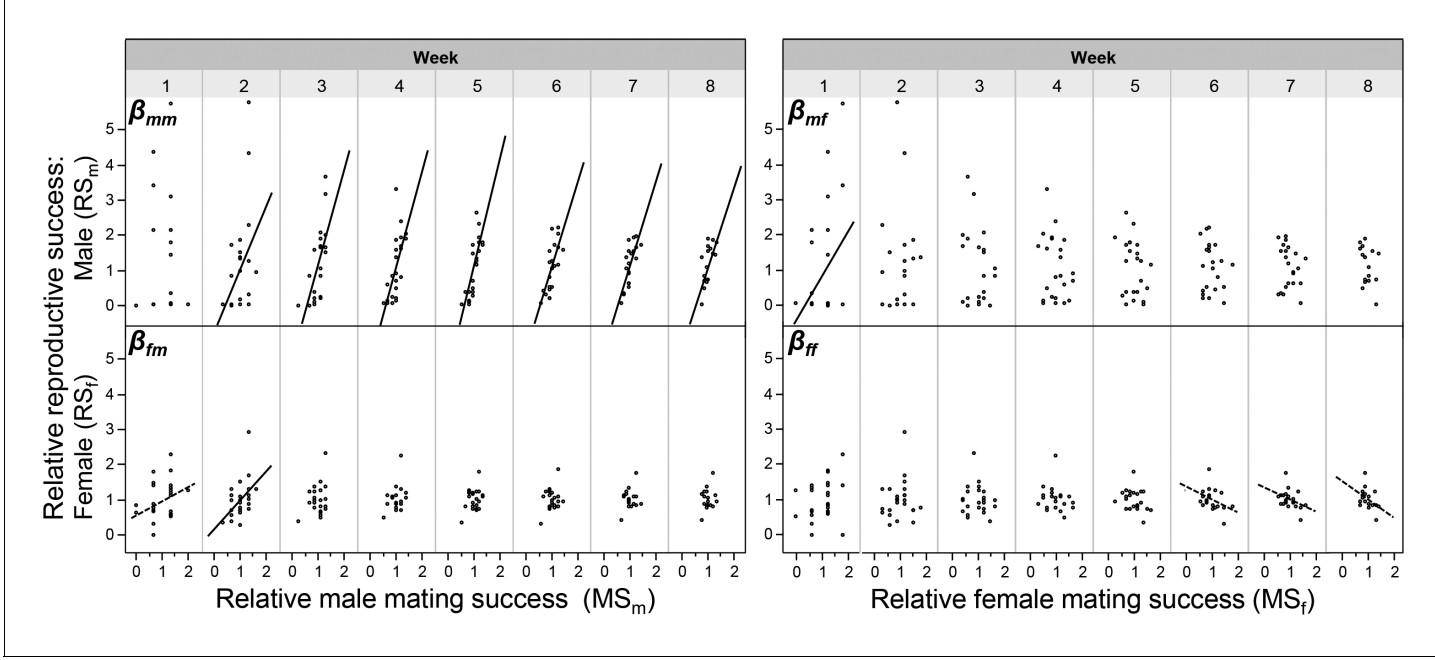

**Figure 2.** The relationships between male and female mating success and reproductive success. The relationships are shown for every week of the experiment. The within-sex and cross-sex sexual selection gradients are based on bivariate regressions of either reproductive success on mating success ($\beta_{mm}$, $\beta_{ff}$, $\beta_{mf}$, $\beta_{fm}$). Significant slopes ($p < 0.05$) are indicated with a solid fitted line, a trend ($p < 0.10$) is indicated with a dashed line. As shown in *Figure 2—source data 3*, the slopes of different weeks do not significantly differ from each other.

The following source data and figure supplement are available for figure 2:

**Source data 1.** The calculated values and their confidence interval (CI) for the opportunity for selection (*I*) and sexual selection (*I*$_s$) for both sexual roles, indicated with subscript m or f, over the weeks.

**Source data 2.** Slope comparisons between weeks for all the significant sexual selection gradients shown in *Figures 2* and *3*.

**Source data 3.** Slope and confidence interval of the correlations between the different sexual selection measures.

**Figure supplement 1.** I-values and gradient values are shown over time, calculated for every week of the experiment.

## Mating mode

During the whole experiment, all the mating interactions were observed. This allowed us to assess whether this species mates fully unilateral or via a form of relaxed reciprocity (*Anthes et al., 2010*). When looking at the relationship between male and female mating success (respectively, MS$_m$ and MS$_f$) in the Multiple partner treatment, no significant correlation emerges (*Figure 4*). We tested this using a GLMM with MS$_m$ and Week as factors, MS$_f$ as dependent variable and focal identity as random factor (MS$_m$: $F_{1,\ 147,6} = 1.314$, $p = 0.254$; Week: $F_{7,\ 144,6} = 12.421$, $p < 0.0001$; Interaction: n.s.). If animals had alternated roles between matings, either with the same or a different partner, this would have resulted in a positive correlation (note that in the Single partner treatment reciprocity is enforced). The statistical significance of the factor Week simply reflects the cumulative nature of these data and reveals that animals keep mating as male and female throughout the experiment, as also illustrated in *Figure 4*. For example, mean cumulative mating success at Week 2 averaged around 3 and at Week 8 around 12 matings for each sexual role.

We also looked at whether there was a clear correlation between male and female reproductive success (respectively, RS$_m$ and RS$_f$), which might be indicative of overall individual quality. While there was a slight correlation in Week 2 ($R = 0.41$, $N = 24$, $p = 0.046$; *Figure 4—figure supplement 1*; which might explain the positive relationship between MS$_m$ and RS$_f$ in the first two weeks, see above), this relationship was absent throughout the rest of the experiment (*Figure 4—figure*

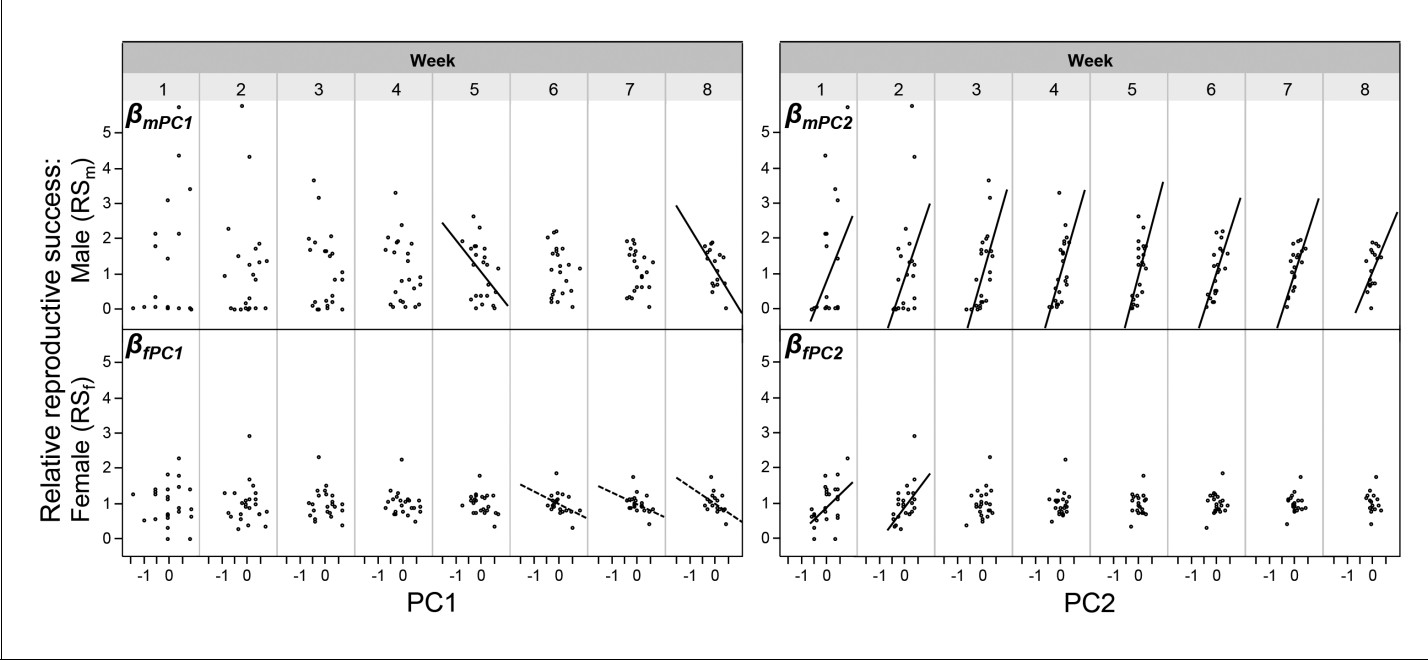

**Figure 3.** The relationships between the PCA values (based on mating success) and reproductive success. The relationships are shown for every week of the experiment. The within-sex and cross-sex gradients are based on bivariate regressions of either reproductive success on principal components ($\beta_{mPC2}$, $\beta_{fPC2}$, $\beta_{mPC1}$, $\beta_{fPC1}$). PC1 represents the sexual bias (the relative difference between $MS_m$ and $MS_f$); PC2 represents the overall mating activity (the correlation component between $MS_m$ and $MS_f$). Significant slopes (p < 0.05) are indicated with a solid fitted line, a trend (p < 0.10) is indicated with a dashed line. As shown in *Figure 2—source data 3*, in italics, the slopes of different weeks do not significantly differ from each other.

The following source data is available for figure 3:

**Source data 1.** Results of the principal component analysis (PCA) for each week of the experiment.

supplement 1). This was also tested using a GLMM with $RS_m$ and Week as factors, $RS_f$ as dependent variable and focal identity as random factor ($RS_m$: $F_{1, 151}$ = 3.115, p = 0.080; Week: $F_{7, 146,8}$ = 28.015, p < 0.0001; Interaction: n.s.). Again, the statistical significance of the factor Week reflects the cumulative nature of these data and reveals that animals keep obtaining male and female reproductive success over the course of the experiment. For example, mean reproductive success at Week 2 was higher for the female than the male role, but these means lie much closer together at Week 8, around 2000 eggs (*Figure 1*). The reason for male and female reproductive success not being equal is found in the fact that selfing occurs at the start of the experiment, which only contributes to $RS_f$.

## Effect of partner availability

In order to evaluate whether repeated mating is beneficial for offspring in this species, we assessed differences in growth, egg laying and hatching at the end of the experiment between the three treatment groups. There was no difference in body size (shell length: ANOVA: $F_{2, 69}$ = 2.385, p = 0.100; body weight: ANOVA: $F_{2, 61}$ = 1.441, p = 0.245) between the focals of the different treatments. Also, we found no difference in number of eggs between the masses produced by the focals of the different treatments at the end of the experiment (ANOVA: $F_{2, 63}$ = 0.155, p = 0.857). Because we did not follow the development of all the egg masses that were laid on the final day of the experiment, we verified whether there was a difference in the number of eggs that we followed per treatment, but this was not the case (ANOVA: $F_{2, 31}$ = 0.168, p = 0.846).

The overall proportion of hatching success and development scored after 14 days differed between the treatments ($\chi^2_2$ = 473.245, p < 0.0001). Using nonparametric multiple comparisons, we found that in the Multiple partners treatment significantly more offspring had reached hatching than

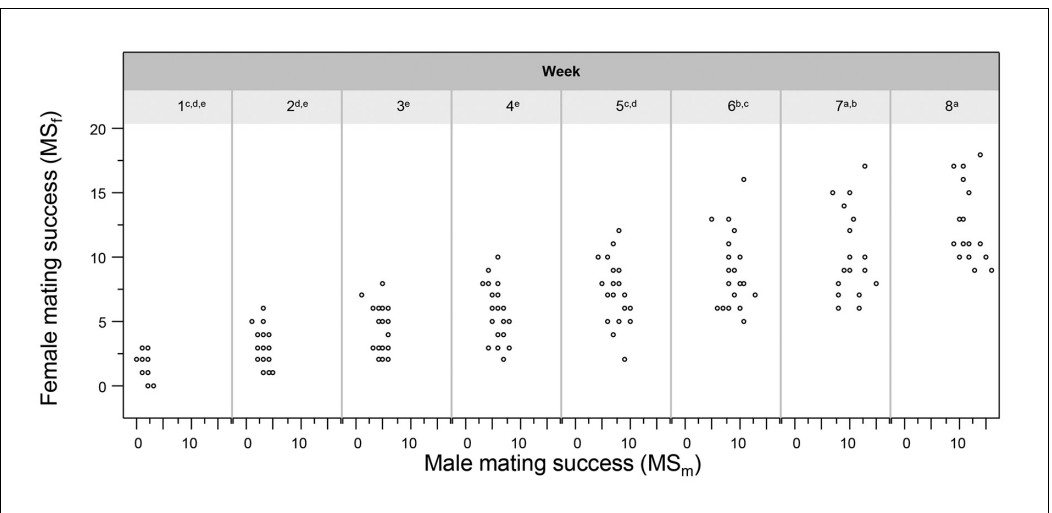

**Figure 4.** The relationship between male and female mating success. The relationship is shown for every week of the experiment. The absence of fitted lines indicates the absence of significance between the individuals' male and female mating success. The superscripted significance letters indicated with the week numbers indicate the Tukey post-hoc differences between weeks.

The following figure supplement is available for figure 4:

**Figure supplement 1.** The relationship between male and female reproductive success.

in the No partner treatment (Wilcoxon: $Z = -2.609$, p = 0.009), while the Single partner treatment did not differ significantly from either treatment (*Figure 5A*). When looking at the proportion of undeveloped offspring, the Single partner and No partner treatments differ significantly from each other (Wilcoxon: $Z = 2.060$, p = 0.039), while the difference between the Multiple partners and No partner treatment shows a trend (Wilcoxon: $Z = 1.714$, p = 0.086; *Figure 5B*).

## Discussion

The study that we present here is the first to look in detail at the effect of repeated mating on reproductive success over time in a simultaneous hermaphrodite. By using a simultaneous hermaphrodite, we could answer several unresolved questions that are relevant for the understanding of sexual selection in general. Firstly, we showed that the potential gain in reproductive success is consistently positive via the male function. Secondly, the existence of cross-sex effects was supported by the sexual selection gradients, emphasizing that their effects are important in the long run. Thirdly, our data clearly showed that sexual selection gradients change over time, which is important for the interpretation of these

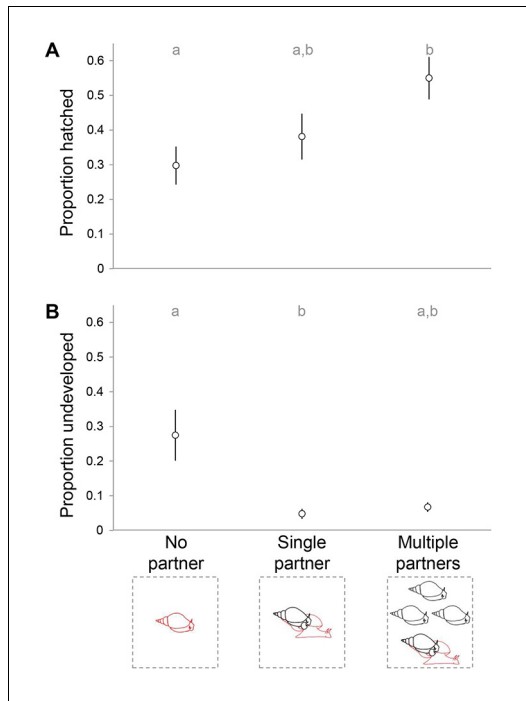

**Figure 5.** Hatching success of eggs collected at the end of the experimental period. The proportion of undeveloped (**A**) and hatched (**B**) eggs is shown for each of the three treatments. The different letters indicate significant differences based on Wilcoxon multiple comparisons (p < 0.05).

values. Fourthly, we showed that in this pond snail, repeated mating was beneficial for the development of offspring. Finally, our data confirmed that the investigated hermaphrodite mates unilateral, without any conditional role alternation. In the following, we will briefly discuss each of these conclusions and place these in the larger context of their general implications for sexual selection.

While it becomes clear from the above that it is very informative to run such an experiment for a longer time, this approach also emphasises that it does complicate the interpretation of the sexual selection measures. In short-term experiments, and (imposed) semelparous situations, one can separate pre- from post-copulatory components of sexual selection by looking at the residual variance in reproductive success, i.e. the variance not explained by mating success (this residual can be explained by post-copulatory processes: *Rose et al., 2013a*, *2013b*; *Pélissié et al., 2014*; *Janicke et al., 2015*). This is no longer straight-forward in a dataset from a longer running experiment where everyone has mated repeatedly with everyone else and sperm from those previous matings is still in storage. So, even though our data show that at the end of the experiment 19% to 52% of the variance in reproductive success (RS) is explained by mating success (MS; which is essentially pre-copulatory), post-copulatory processes, such as sperm storage and sperm competition, have had time to act on reproductive success (and are thus partially included). Hence, one can still attribute the remaining variance to post-copulatory processes, but this no longer captures all this variance. For example, age now becomes a confounding factor and cannot be fully excluded. These considerations are of biologically relevance because they apply to any species, separate sexed or hermaphroditic, that mates multiple times over an extended reproductive period and stores sperm before producing eggs. The only studies that have been able to deal with this issue partly (though not the factor age), so far, are the ones by *Pélissié et al. (2014)* and *Turnell and Shaw (2015)*. They achieved this by taking mating order into account, which was facilitated by the short duration of their experiments, and their findings largely corroborate what we observe early on in our experiment. To also be able to take the effect of age into account, a similar experiment to the one presented here would need to be run in which the virgin snails (and their mating partners) have different standardized ages at the start of the eight-week experiment.

Our data reveal that repeated mating, especially with several different partners is beneficial for the hatching success of the offspring. This is especially true when the Multiple partner treatment is compared to the No partner treatment. Hence, this could also suggest that self fertilization, as occurred in the No partner treatment, is detrimental to offspring. Similar effects of selfing have been found for many snail species (e.g., *Escobar et al., 2011*), and are corroborated by the higher number of undeveloped eggs in that treatment. Note that in this case the non-significant difference between the No partner and Multiple partners treatments, although they seem very different, is due to both sample size and the non-parametric test that needed to be used. Interestingly, results so far had not indicated very clear negative effects of selfing for *L. stagnalis* (*Escobar et al., 2011*; *Puurtinen et al., 2007*), so the fact that we do observe it here may reflect that we looked at the development of offspring in more detail, or that such effects only become apparent in the long run. In addition, while the Single partner treatment differs from neither of the other treatments, it is suggestive of a gradient in which repeated mating may also be beneficial for hatching success when it occurs with the same partner (but to properly answer this question, a Single copulation treatment would need to be included in a follow-up study). Notwithstanding, our data are the first to show that multiple mating does offer a benefit for the development of eggs of *L. stagnalis*, and are supported by previous work that showed that such eggs are heavier (*Hoffer et al., 2012*). These findings are also in line with work on other species (e.g., *Callosobruchus maculatus*: *Power and Holman, 2014*). It remains to be investigated whether this effect on offspring has any further implications on *Lymnaea*'s growth and survival.

Multiple mating has already been shown to preferentially occur with different partners (*Koene and Ter Maat, 2007*) and behavioural studies imply that this species does not alternate mating roles conditionally (i.e., they can swap sexual roles within one mating interaction, but this is not obligatory: *Koene and Ter Maat, 2005*). This is of importance since it has implications for whether sexual selection can act independently on the two sexual roles of a hermaphrodite (*Anthes et al., 2010*; see also *Arnold, 1994*). Given that our current data reveal no relation between $MS_m$ and $MS_f$, this supports that the mating mode of this species is unilateral (no exchange of sex roles). Hence, male and female strategies can probably be optimized independently in this species. The lack of a clear correlation between $RS_m$ and $RS_f$ also indicates that an individual successful in the

male role is not necessarily successful in the female role. The latter is also supported by our finding that reproductive success can be gained constantly via the male function, as shown by the significant male sexual selection gradients ($\beta_{mm}$) throughout the experiment.

Given that this mating system seems largely male-driven, we expected to also see cross-sex effects. These effects should become visible as negative gradients when looking at male or female reproductive success (respectively, $\beta_{mf}$ or $\beta_{mPC1}$ and $\beta_{fm}$ or $\beta_{fPC1}$). Such negative effects emerged more clearly from the PCA approach, and only near the end of the experimental period ($\beta_{fPC1}$). This may explain why such cross-sex effects were not found in earlier studies that lasted only for several days (*Anthes et al., 2010*; *Pélissié et al., 2012*; *Pélissié et al., 2014*). Moreover, our results highlight that cross-sex effects do have an impact on the reproductive success of this hermaphroditic animal. The cross-sex effects that we observe can potentially be explained by the reported negative effects of seminal fluid proteins on both sexual functions (*Koene et al., 2010*; *Nakadera et al., 2014b*), but this remains to be tested directly. Interestingly, we did not find any indication for a trade-off between investing in the two sexes, which would have resulted in a negative $\beta_{fm}$ (although a trend is seen in the last weeks of the experiment for $\beta_{fPC1}$). Based on sex allocation theory, one might have expected that increased mating success as a male trades off with female reproductive success ($\beta_{fm}$), or vice versa ($\beta_{mf}$; *Anthes et al., 2010*; *Charnov, 1979*; *Schärer, 2009*).

As mentioned above, other studies on different simultaneous hermaphrodites did not find any cross-sex effects, while they did find that the mating system is mainly driven by the male function (*B. glabrata*: *Anthes et al., 2010* and *P. acuta*: *Pélissié et al., 2012*, *Pélissié et al., 2014*; but see *Janicke et al., 2015*). An important difference between those studies and our study is that while they followed individuals for 3 to 5 days, we followed them for 56 days (8 weeks). This also allowed us to evaluate the effect of the chosen time frame on the results of such studies. Our data showed that from the start, the main effect, the significant $\beta_{mm}$, is already captured (also with the PCA approach: $\beta_{mPC2}$). This relationship became statistically stronger over time. In contrast, the negative effects only emerged much later into the experiment, which indicates that such effects on reproductive success may only be detectable in the longer run. Hence, short term experiments only take a snap shot of reproductive success (of virgins).

To conclude, we showed that *L. stagnalis* has a unilateral mating system and that sperm donors gain most reproductive success from repeated mating, even though they seem to lose some reproductive success in the long run (a cross-sex effect from mating frequently as a female). Our data also do reveal that these sperm recipients benefit indirectly from repeated mating, since their egg development and hatching success is higher. Our experiment therefore showed that the experimental time frame is very important for the quantification and interpretation of sexual selection measures, an insight that applies to any mating system with multiple mating.

## Materials and methods

### Study species

The basommatophoran pond snail *L. stagnalis* is a species common to the Holarctic region, and resides in ponds, ditches, and lakes. At the mass culture facility at VU University, a laboratory population of *L. stagnalis* has been maintained on running, low-copper water for more than 50 years (*Van Der Steen et al., 1969*). Snails are kept under a 12L:12D photoperiod. Each month, egg masses laid within a 24 hr time frame are raised to become the next generation. Snails are alternatingly fed fish food flakes (TetraPhyll GmbH.) and broad leaf lettuce. At a shell length of about 18 mm, around the age of two months, individuals begin to copulate, soon followed by production of egg masses.

*L. stagnalis* has a mixed mating system with high outcrossing rates despite low self-fertilization depression (*Nakadera et al., 2014a*, *Nakadera et al., 2017*; *Puurtinen et al., 2007*; *Cain, 1956*; *Coutellec and Caquet, 2011*; *Koene et al., 2009*). A single copulation interaction is unilateral, meaning that one partner performs the male role and the other the female role. After an initial copulation role-alternation can take place (*Koene and Ter Maat, 2005*) and also chain-copulations can be observed in groups. Mating rates increase with population density (*Koene and Ter Maat, 2007*) and copulation can be easily observed in the laboratory (*Van Duivenboden and Ter Maat, 1988*; *De Boer et al., 1996*). *L. stagnalis* is a promiscuous species and can store received sperm (allosperm)

for about 2 months (62 days: *Nakadera et al., 2014a*) and use it seemingly random when fertilizing eggs (*Koene et al., 2009*). Pond snails are highly fecund and usually lay between 100–300 eggs per week in 1–4 egg masses, depending on mating conditions (*Hoffer et al., 2012*; *Van Duivenboden et al., 1985*).

## Experimental design

Two hundred immature snails (shell length < 15 mm), that hatched from multiple egg masses laid on the same day (±24 hr), were collected from our mass culture. They were individually housed in perforated plastic jars in a laminar-flow basin (20 ± 1°C) to let them reach maturity. A bee tag was glued to their shell for identification purposes. For the duration of the maturation period and the experiment, 19.6 cm$^2$ of lettuce was provided daily per animal. During this maturation period, a clean jar was provided weekly, and egg laying capability was checked. At 14 weeks after hatching (shell length ~30 mm) all virgins were confirmed to be laying self-fertilized eggs. After this confirmation, all snails were sedated with 50 mM $MgCl_2$ and, using fine surgical scissors (World Precision Instruments, Inc., Saratosa, USA), a small part of their foot was cut off for genotyping purposes (see next section). The quantification of mating success (MS) and reproductive success (RS) started when animals - still virgin - were 110 days old, at which time they had had plenty of time to fully recover from tissue sampling.

The experiment included three treatments: Multiple partners (25 groups of 5 snails each), Single partner (25 groups of 2 snails each), and No partner (25 single snails that remained virgins). The Single partner and No partner treatments were included in the experiment to test for potential effects of repeated mating with different partners in the Multiple partners treatment. For the snails in groups, we made sure that the focal had a microsatellite genotype (see Genotyping protocol) that could be unequivocally distinguished from the other snails in its group. The genotypes of the non-focal individuals necessarily overlapped because we found three alleles at this locus within our experimental population.

During eight weeks, for each treatment mating activity was observed twice a week for 7 hr on the first and fourth day of each week (*Figure 1*). The snails only had access to mating partners during these 7 hr mating trials, when they were together in a jar. During this time, the volume of water per individual was set at 100 ml, so that snail density was equal among treatments (500 ml for Multiple partner groups, 200 ml for Single partner pairs, and 100 ml for No partner virgins). The rest of the time all snails were kept in their isolation jars (*Figure 1A*). Within all groups, copulation and mating role was noted for each individual (175 snails in total all of which mated more than once in the male and female role), hence we had complete observational data based on which we could calculate each individual's MS. We used cumulative number of matings, not mates, because this species mates frequently and within the first weeks all individuals have already mated at least once with each group member in both sexual roles, hence maximizing number of mates. Thus, due to the restricted number of different mates available, Bateman gradients could not be quantified as these are defined based on mate identity. By looking at mating frequency we thus determined sexual selection gradients rather than Bateman gradients sensu stricto (see also *Anthes et al., 2010*; *Collet et al., 2014*; *Fritzsche and Arnqvis, 2013*; *Marie-Orleach et al., 2016*).

Egg masses laid during mating trials were removed from the container and placed in the isolation jar of the mother. Each week, egg masses were collected from the isolation jars, measured to the nearest 0.5 mm, and placed in 10 ml vials, one for every individual. The egg numbers of 24 random egg masses per treatment were counted, making it possible to estimate the number of eggs in an egg mass of length x for each week. After counting, egg masses were returned to their vials and then freeze-dried. The total dry weight of all the egg masses was determined on a microbalance (type 1712 MP8, Sartorius) to the nearest 0.01 mg. In the first, second, fourth, sixth and eighth week, one egg mass per individual (if any) from the Multiple partners treatment was allowed to develop until the embryo was large enough for genotyping (at 9–10 days after egg laying). Then, these masses were freeze dried, weighed, and stored at −20°C until offspring genotyping.

For all snails, shell length and body weight were measured on day 15 and day 57 after the start of the behavioural observations. Growth in *L. stagnalis* follows a sigmoid curve, and the experimental period started while the snails had entered the asymptotic phase, thus restricting potential budget effects. Egg masses laid on day 57 (the day after the eighth week; the end of the experiment) were placed in Petri dishes with 15 ml of water each, which was refreshed every other day for two weeks.

After 14 days, for each egg mass the number of undeveloped eggs, early embryos, late embryos and hatchlings were counted under a stereo microscope.

## Genotyping protocol

For genotyping the experimental animals, total genomic DNA was extracted by crushing tissue samples in 100 µl 50 mM NaOH in a 1.5 ml vial, vortexed and left standing for 10 min. After digestion of the connective tissue the solution was neutralized by adding 10 µl 1 M TRIS-HCl of pH 8.0 (protocol adapted from *Meeker et al., 2007*). After centrifugation at 14000 rpm for 10 min, the supernatant was transferred to a clean vial. The precipitate containing the tissue debris was discarded.

PCR amplification of the A16 microsatellite locus (*Knott et al., 2003*; please note that the reverse primer is displayed in 3′−5′ orientation in that publication) was performed in 25 µl reaction mixture containing 5.0 µl of 5x PCR buffer, 1.5 µl of 25 mM MgCl$_2$, 2.0 µl of 10 mM dNTP's, 1 µl of the 5 µM forward and reverse primer each, 0.2 µl GO-taq polymerase (Promega), plus 0.02 µl proofreading polymerase (pfu, Promega) and 9.3 µl H$_2$O (Sigma). Lastly, 5 µl of genomic DNA sample was added to the reaction mixture. The PCR amplification protocol consisted of an initial denaturation at 95°C for 5 min, followed by 35 cycles of 95°C for 15 s, 55°C for 45 s, and 72°C for 60 s, with a final extension period of 72°C for 5 min in a thermocycler (MJ Research PTC-100). A volume of 16 µl amplification product was added to 4 µl loading dye (Elchrom Scientific) which was then denatured at 95°C for 5 min, and chilled on ice. Spreadex EL 600 Wide Mini Gels (S-2 × 25 slots, Elchrom Scientific) were submerged in a buffer solution (55°C, 0.8x TAE), and slots were carefully filled with PCR product, including one slot for each half gel for a 250 bp DNA ladder. Electrophoresis was performed at 120 V for 165 min, with a second loading PCR product after 45 min. Gels were stained in 150 ml 0.25x TAE buffer containing 15 µl Syber Gold for 45 min. All gels were photographed and snails were visually genotyped by two persons, without inconsistencies.

Genotyping of offspring was performed on single eggs. Because the high fecundity of *L. stagnalis* (~100–300 eggs per week per individual) ruled out complete genotyping of all offspring, we genotyped between 10 and 24 (16 on average) randomly selected offspring per developed egg mass of focal and non-focal individuals (see also Experimental design). Sixteen embryonic snails, recognized by their dark colour, were removed randomly from an egg mass and put singly in a well of a 96-wells PCR-plate. When the plate was full (6 × 16 offspring) the tissue was crushed in 50 µl 50 mM NaOH, incubated at room temperature for 10 min, and neutralized with 5 µl 1 M TRIS-HCl of pH 8. After centrifugation, supernatant was either stored at −20°C or used for amplification directly. The PCR and electrophoresis protocol was identical to the one mentioned above.

## Reproductive success

For all focal individuals, reproductive success in the female role (RS$_f$) was expressed as the number of eggs produced by the focal, and was calculated on a weekly basis. Male reproductive success (RS$_m$) of the focals was estimated based on the genotyping data of the A16 microsatellite for the random subset of eggs as described above (see Genotyping protocol). Because we had incomplete paternity sampling (i.e., not all eggs from each individual were genotyped; see also *Mobley and Jones, 2013*), we used the observed paternity and overall proportion of female matings of each mate within each group to estimate paternity for the non-genotyped egg masses. We entered this proportion and the actual number of fathered offspring (determined by genotyping) into a generalized linear model (GLM). We used a binomial distribution and a logit link function (logistic regression), corrected for overdispersion and used the number of genotyped offspring as a weighing factor (GLM fit: $\chi^2_1 = 64.156$, $p < 0.0001$). This predicted paternity share was then multiplied by the total number of offspring produced by the mating partners in each group, thus resulting in an estimate of RS$_m$ for each focal (see *Pélissié et al., 2012*). All statistical procedures were performed using JMP 9.0.0 (SAS).

## Quantifications of sexual selection

Due to different limits in terms of their gamete production (as explained above), variance in the reproductive success of males is expected to be larger than that of females and can be captured in the variance measure *I*. This measure is defined as the standardized variance in relative reproductive success and its value is indicative of the opportunity for selection (*Gowaty et al., 2003*;

*Jones, 2009*). The opportunity for sexual selection, which is defined as the standardized variance in relative mating success, is captured in the variance measure $I_s$ (*Arnold, 1994*; *Anthes et al., 2010*). In contrast to such opportunity values, a real measure of sexual selection can be obtained by looking at the relationship between mating success and reproductive success, which generally results in a steeper regression line for males than females. The slope of such a linear regression line is referred to as the Bateman or sexual selection gradient ($\beta$; e.g., *Arnold and Duvall, 1994*; *Anthes et al., 2010*, *Anthes et al., 2016*).

We first calculated the opportunity for (overall) selection, $I$, by dividing RS's variance by its squared mean. Likewise, we calculate the opportunity for sexual selection, $I_s$, by dividing MS's variance by its squared mean. Given that we are dealing with a simultaneous hermaphrodite, these can be calculated both for the male ($I_m$, $I_{sm}$) and female ($I_f$, $I_{sf}$) role (see *Lorenzi and Sella, 2008*; *Shuster and Wade, 2003*). Subsequently, we calculated the other important, and often used, measure of sexual selection, the sexual selection gradient ($\beta$). As already explained in the Results section, this is the linear least-squares regression slope of sex-specific relative RS on sex-specific relative MS (*Jones, 2009*; *Klug et al., 2010*). To deal with the non-independence of male and female reproduction in these hermaphrodites, we used a multiple regression with $MS_m$ and $MS_f$ as explanatory variables on, respectively, $RS_f$ and $RS_m$.

For simultaneous hermaphrodites, depending on their mating system, male and female mating success may not be fully independent. Even in unilaterally mating species, reciprocity in mating (i.e., playing both roles in a mating encounter) could make a multiple regression analysis statistically fragile (*Wacker et al., 2014*). We followed *Anthes et al. (2010)* suggestion to replace $MS_m$ and $MS_f$ by their principal components (PC) to cope with this potential problem. This approach results in two completely independent new variables (PC1 and PC2) that represent overall mating activity and the sex bias in mating, not necessarily in that order.

Finally, we investigated the effect of time on the above measures by comparing the measures of sexual selection over time in the experiment using a GLMM. So far, experiments were generally only performed over a short time frame, as pointed out earlier; e.g. *Pélissié et al., 2012*, *Pélissié et al., 2014*; *Anthes et al., 2010*; *Collet et al., 2012*; *Rose et al., 2013a;* but see *Turnell and Shaw, 2015*).

## Acknowledgements

We are thankful for the assistance of EM Swart, R van Oosten, ZV Zizzari, and Y Nakadera during this experiment, and thank C Popelier for maintaining our snail cultures. Also, we thank the participants of the SHAM (Simultaneous Hermaphrodite Animals Meeting 2009, 2010) and SHOW (Simultaneous Hermaphroditic Organisms Workshop, 2011–2013) meetings for fruitful discussions. A particular thanks goes out to N Anthes, P David, I Häderer, T Janicke, M Lodi, L Marie-Orleach, Y Nakadera, E Noel, B Pélissié, L Schärer, EM Swart, D Vizoso and ZV Zizzari for in depth discussions (at different stages) about the execution, analysis and interpretation of this work. Two anonymous reviewers provided constructive comments and suggestions, for which we are grateful. This research was financed by the Research Council for Earth and Life Sciences (ALW) with financial aid from the Netherlands Organization for Scientific Research (NWO) via a grant to JMK (grant number 816.01.009).

## Additional information

### Funding

| Funder | Grant reference number | Author |
| --- | --- | --- |
| Nederlandse Organisatie voor Wetenschappelijk Onderzoek | 816.01.009 | Joris M Koene |

The funders had no role in study design, data collection and interpretation, or the decision to submit the work for publication.

## Author contributions

JNAH, Conceptualization, Data curation, Formal analysis, Investigation, Visualization, Methodology, Writing—original draft, Writing—review and editing; JM, Supervision, Methodology, Writing—original draft; JE, Resources, Supervision, Methodology, Writing—original draft, Project administration, Writing—review and editing; JMK, Conceptualization, Resources, Data curation, Software, Formal analysis, Supervision, Funding acquisition, Validation, Investigation, Visualization, Methodology, Writing—original draft, Project administration, Writing—review and editing

## Author ORCIDs

Joris M Koene, http://orcid.org/0000-0001-8188-3439

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
