## [Decision Letter]

Thank you for submitting your article "Sexual selection gradients change over time in a simultaneous hermaphrodite" for consideration by *eLife*. Your article has been favorably evaluated by Ian Baldwim (Senior Editor) and three reviewers, one of whom is a member of our Board of Reviewing Editors. The following individual involved in review of your submission has agreed to reveal his identity: Tim Janicke (Reviewer #3).

The reviewers have discussed the reviews with one another and the Reviewing Editor has drafted this decision to help you prepare a revised submission.

Summary:

The manuscript by Hoffer et al. tests the Darwin-Bateman paradigm that sexual selection acts more strongly on males than females and uses a simultaneously hermaphroditic pond snail (*Lymnaea stagnalis*) as a model system to successfully test this hypothesis. Previously it was demonstrated that *Lymnaea stagnalis* perform more inseminations in larger groups and prefer to inseminate novel over familiar partners. However, the evolutionary benefit to this phenomenon has not been tested empirically previously, because so far sexual selection was studied during short time windows in a given experimental setting. The authors now show that during mating observations over 8 weeks, comprised of 16 mating trials and 8 times of egg collections, males gain more in reproductive success from repeated mating than females do, demonstrating positive sexual selection gradients in male. This demonstrates how time affects the estimated strength of sexual selection. Finally, the authors conclude that the offspring generation of this simultaneous hermaphrodite improves due to mating with multiple partners. In conclusion, this study is a very timely and important contribution to the literature that will be of interest to biologist working on sexual selection in general.

Essential revisions:

1) Discussion: "Secondly, the expected (negative) effects of seminal fluid proteins were supported by the sexual selection gradients, emphasizing that their effects are important in the long run." The experiment did not explicitly test the role of seminal fluid proteins and, therefore, this sentence and the associated discussion throughout the manuscript represents an over statement. In particular, because the mechanism is not corroborated from the statistics in a direct manner. After this concern is fully addressed the paper would still represent a significant advancement in our understanding of sexual selection, in particular the role of the Darwin-Bateman paradigm is of broad interest.

2) To describe relationships between male and female mating success and reproductive success, bivariate regressions are calculated and plotted (Figure 2). To better describe the observed relationships, please calculate the statistical difference (e.g. by δ test) between individual slopes among different weeks and provide the results of the analysis as a supplementary table. Furthermore, since the interpretation of the results is based on the statistical significance of the bivariate regression, please provide the values for the measurements and discuss.

3) Please perform similar calculations and analysis (as in 2) for the PCA analysis in Figure 3. Although the rationale for reducing the data dimensionality is given, please consider that it is challenging to interpret the data as even significant bivariate regression of individual components do not necessarily represent the biological relationship of the individual parameters. Further, whereas the PCA analysis in Figure 3 is used to support the results in Figure 2, it seems the figures might reveal that β_fPC_1 and β_fPC2_ are similar to β_ff_ and β_fm_, respectively.

4) Please discuss how the variation in selection gradient over time can be biased by additional factors. For example, how might the sexual selection gradient depend on accumulated mating or aging. Would the same mating and reproductive success be observed in the snails if mating is performed 8 weeks later than in the original experimental group?

5) Please better organize the Results, Discussion, and Materials and methods in the corresponding sections. Jargon can be replaced, reduced and explained, consider using the definitions in Jones et al. (2000) and clarifying equations like Arnolds & Duvall (1994). The figures axis should be labeled with full names instead of variable names (for example 'Reproductive Success' instead of 'RS'). Please improve the overall flow, if references to earlier sections are essential, introduce sub-headings that can be referenced more easily. However, ideally the flow of the text is such that cross referencing to different sections is not needed.

6) Improve the statistical rigor. The manuscript offers limited statistical support for the conclusion that the strength of sexual selection depends on the tested time window. For the male sex function, selection may not change over time as indicated by a non-significant week x RS(male) (subsection “Quantification of sexual selection”, second paragraph). However, such a statistical test is not provided for the female sex function, which is supposed to change over time. The main statistical support for a temporal effect is given for the cross-sex effect of RS(female) on MS(male). The same is true for the temporal patterns in I and Is. Even though Figure 2—figure supplement 1 suggests a strong decline in the opportunity for (sexual) selection, there is no statistical assessment of such an effect (and the same is true for the postulated sex difference). Please provide a more explicit statistical analysis for the effect of week on the sexual selection metrics. E.g. a test for the week x RS(female) interaction on MS(female). Also consider using bootstrapping to obtain confidence intervals for all tested sexual selection metrics (i.e. I, Is, β[mm], β[ff]). E.g. this can be performed in R or the freely available software package BATMANATER (Jones 2015).

7) Clarify the benefits of multiple mating for offspring development. There seems to be a significant increased hatching success in the 'multiple partners' treatment compared to the 'no partner' treatment, but no difference between the 'single partner' and the other two treatments. If there would be an effect of repeated matings (with the same partner), we might also expect a difference between the 'single' and the 'no' partner treatment, and if there was an effect of polyandry (i.e. mating with different partners) we might expect a difference between the 'multiple' and the 'single' partner treatment. The data may also indicate that mating versus self-fertilization in the 'no partner' treatment (but not necessarily multiple mating, is beneficial for offspring development (Discussion, third paragraph). Further, if there is a negative effect of self-fertilisation (i.e. inbreeding depression), one might also expect a difference between the 'single partner' and the 'no partner' treatment. But here the question arises whether all pairs did actually copulate in the 'single partner' treatment. Based on work on another hermaphroditic freshwater snail, it was noted that not all individuals copulate when kept in pairs, leading to self-fertilization, and therefore lowering hatching success in pairs compared to groups. Therefore, we suggest that the authors re-evaluate their findings regarding the benefit of multiple mating or provide a more detailed explanation of why they believe that multiple mating (i.e. mating with different partners, Discussion, third paragraph) is beneficial in their model system given the presented data.

---

## [Author Response]

*Essential revisions:*

*1) Discussion: "Secondly, the expected (negative) effects of seminal fluid proteins were supported by the sexual selection gradients, emphasizing that their effects are important in the long run." The experiment did not explicitly test the role of seminal fluid proteins and, therefore, this sentence and the associated discussion throughout the manuscript represents an over statement. In particular, because the mechanism is not corroborated from the statistics in a direct manner. After this concern is fully addressed the paper would still represent a significant advancement in our understanding of sexual selection, in particular the role of the Darwin-Bateman paradigm is of broad interest.*

You are right that this was not explicitly tested. What we have done in response is to eliminate any reference to possible effects of seminal fluid proteins being tested (and/or text implying a mechanism for what was observed in this respect): – in the Abstract, the reference to Seminal fluid protein (SFP) effects has been removed;In the Discussion, we have removed the “expected negative effects of SFPs” and replaced this by saying that we do find support for cross-sex effects. – The fifth paragraph of the Discussion has now been edited to refer to cross-sex effects, all of the direct reference to this being due to SFPs has been deleted, except for one sentence pointing out that it would be useful to test his prediction now. In the sixth paragraph of the Discussion, that this could be due to SFPs has been deleted. – In the concluding paragraph, mention of SFPs has also been removed.. As becomes clear from the above, we now do address the cross-sex effects without interpreting them as potential evidence for being the result of SFPs. The only sentence remaining that alludes to this as a possible explanation, is one in the Discussion (fifth paragraph) that points out that the observed cross-sex effect can potentially be explained by the previously-demonstrated effects of seminal fluid proteins in this species. We sincerely hope that you agree with the way we have revised the manuscript in this respect.

*2) To describe relationships between male and female mating success and reproductive success, bivariate regressions are calculated and plotted (Figure 2). To better describe the observed relationships, please calculate the statistical difference (e.g. by δ test) between individual slopes among different weeks and provide the results of the analysis as a supplementary table. Furthermore, since the interpretation of the results is based on the statistical significance of the bivariate regression, please provide the values for the measurements and discuss.*

Thank you very much for this suggestion. We have now included an additional supplementary file ([Supplementary-material SD2-data]). This table reports the actual slope values, and reports the statistical comparison between the significant slopes indicating that these do not differ between weeks. The test statistics for these comparisons are indicated in the same table and referred to in the third paragraph of the subsection “Quantification of sexual selection”.

*3) Please perform similar calculations and analysis (as in 2) for the PCA analysis in Figure 3. Although the rationale for reducing the data dimensionality is given, please consider that it is challenging to interpret the data as even significant bivariate regression of individual components do not necessarily represent the biological relationship of the individual parameters. Further, whereas the PCA analysis in Figure 3 is used to support the results in Figure 2, it seems the figures might reveal that β_fPC1_ and β_fPC2_ are similar to β_ff_ and β_fm_, respectively.*

As in response to point 2, we have added these comparisons in the same [Supplementary-material SD2-data] and referred to in the last paragraph of the subsection “Quantification of sexual selection”. As for your further advice about the interpretation, as we indeed already intended to point out in the text (now in the aforementioned paragraph), β_fPC1_ and β_fPC2_ are similar to β_ff_ and β_fm_. As you correctly spotted, these were inverted in the following corrected sentence:

“In other words, the slope β_mPC2_ represents β_mm_ and β_fPC2_ represents β_fm_, and the cross-sex effects are seen in β_mPC1_ (=β_mf_) and β_fPC1_ (= _βff;_ Anthes et al. 2010)”. We agree with you and hope this becomes clear from the text now.

*4) Please discuss how the variation in selection gradient over time can be biased by additional factors. For example, how might the sexual selection gradient depend on accumulated mating or aging. Would the same mating and reproductive success be observed in the snails if mating is performed 8 weeks later than in the original experimental group?*

If one were to start with virgin snails that were eight weeks older, the expectation would probably be that one would see the same pattern develop over time. After all, in the initial week(s) matings accumulate. However, this remains to be tested and we now point this out in the second paragraph of the Discussion, also including a suggestion for how this would need to be tested and addressed.

*5) Please better organize the Results, Discussion, and Materials and methods in the corresponding sections. Jargon can be replaced, reduced and explained, consider using the definitions in Jones et al. (2000) and clarifying equations like Arnolds & Duvall (1994). The figures axis should be labeled with full names instead of variable names (for example 'Reproductive Success' instead of 'RS'). Please improve the overall flow, if references to earlier sections are essential, introduce sub-headings that can be referenced more easily. However, ideally the flow of the text is such that cross referencing to different sections is not needed.*

To keep this paper accessible, we have refrained from equations, but we have tried to reduce the jargon and to move the explanation of the sexual selection gradients to a more logical place (from Methods to Results). You will find this moved and edited section in the second paragraph of the subsection “Quantification of sexual selection”. I hope that you agree that this improved the flow and readability of the text.

Thank you very much for the suggestion for clarifying the labels of the figures (notable Figure 2 and Figure 3) as you will see we have done so everywhere where we thought this was necessary. However, please feel free to let us now if you think we have missed out on important labeling. We have not removed all the abbreviations (e.g. RS and MS) from the main text, we assumed that these were not the abbreviations and jargon you were referring to. But if you think this will further improve the flow of the paper, we can do so (although this will significantly increase the overall word count).

*6) Improve the statistical rigor. The manuscript offers limited statistical support for the conclusion that the strength of sexual selection depends on the tested time window. For the male sex function, selection may not change over time as indicated by a non-significant week x RS(male) (subsection “Quantification of sexual selection”, second paragraph). However, such a statistical test is not provided for the female sex function, which is supposed to change over time. The main statistical support for a temporal effect is given for the cross-sex effect of RS(female) on MS(male). The same is true for the temporal patterns in I and Is. Even though Figure 2—figure supplement 1 suggests a strong decline in the opportunity for (sexual) selection, there is no statistical assessment of such an effect (and the same is true for the postulated sex difference). Please provide a more explicit statistical analysis for the effect of week on the sexual selection metrics. E.g. a test for the week x RS(female) interaction on MS(female). Also consider using bootstrapping to obtain confidence intervals for all tested sexual selection metrics (i.e. I, Is, β[mm], β[ff]). E.g. this can be performed in R or the freely available software package BATMANATER (Jones 2015).*

A) The test of the interaction between week and MS(female) on RS(female) was indeed missing from the text. Thank you very much for pointing this out. As we now indicate in the third paragraph of the subsection “Quantification of sexual selection”, this interaction was not significant even though the effect of female mating success on female reproductive success seems to be mainly caused by the negative trend lines in the last three weeks. While editing, we also realized that the explanation of the other interaction may also not have been entirely clear, so we have now tried to improve the phrasing as follows: “For the cross sex effect β_fm_ the model revealed a significant effect of relative MS_m_ on relative RS_f_ (*F_1, 140,5_*= 13.523; *P* = 0.0003), which seem due to the initial positive correlation (in Week 1 and 2; Figure 2 and Figure 2—figure supplement 1), even though the interaction term Week*relative MS_m_ was not significant.”

So, also here, even though the sexual selection gradients (and the significance of their fits) changes over time, this particular model does not reveal this in its interaction term.

B) As for the addition of confidence intervals, we have added tables containing the CI’s for the Β and I values. These are now presented in two supplementary files (1 and 2) and referred to in the first and third paragraphs of the subsection “Quantification of sexual selection”).

*7) Clarify the benefits of multiple mating for offspring development. There seems to be a significant increased hatching success in the 'multiple partners' treatment compared to the 'no partner' treatment, but no difference between the 'single partner' and the other two treatments. If there would be an effect of repeated matings (with the same partner), we might also expect a difference between the 'single' and the 'no' partner treatment, and if there was an effect of polyandry (i.e. mating with different partners) we might expect a difference between the 'multiple' and the 'single' partner treatment. The data may also indicate that mating versus self-fertilization in the 'no partner' treatment (but not necessarily multiple mating, is beneficial for offspring development (Discussion, third paragraph). Further, if there is a negative effect of self-fertilisation (i.e. inbreeding depression), one might also expect a difference between the 'single partner' and the 'no partner' treatment. But here the question arises whether all pairs did actually copulate in the 'single partner' treatment. Based on work on another hermaphroditic freshwater snail, it was noted that not all individuals copulate when kept in pairs, leading to self-fertilization, and therefore lowering hatching success in pairs compared to groups. Therefore, we suggest that the authors re-evaluate their findings regarding the benefit of multiple mating or provide a more detailed explanation of why they believe that multiple mating (i.e. mating with different partners, Discussion, third paragraph) is beneficial in their model system given the presented data.*

We agree with your assessment and have tried to reflect this in the edited text. As you will see below, we have made several changes to address the different issues that you raised.

A) Mating: We now point out more clearly that the animals all mated repeatedly (we observed every mating) in both sexual roles. As we already pointed out: “For example, mean cumulative mating success at Week 2 averaged around 3 and at Week 8 around 12 matings for each sexual role.” And: “within the first weeks all individuals have already mated at least once with each group member in both sexual roles,”

B) Treatment differences: We now added that the following should be considered: “Note that in this case the non-significant difference between the No partner and Multiple partners treatments, although they seem very different, is due to both sample size and the non-parametric test that needed to be used.”

C) Selfing: This is an interesting topic that clearly warrants further investigation. As we now point out that to properly address the reported differences, this requires further testing: “but to properly answer this question, a Single copulation treatment would need to be included in a follow-up study”.